# Lignosulfonate Rapidly Inactivates Human Immunodeficiency and Herpes Simplex Viruses

**DOI:** 10.3390/medicines8100056

**Published:** 2021-10-03

**Authors:** Kunihiko Fukuchi, Takuro Koshikawa, Daisuke Asai, Megumi Inomata, Hiroshi Sakagami, Hiromu Takemura, Taisei Kanamoto, Hikaru Aimi, Yuji Kikkawa

**Affiliations:** 1Graduate School of Health Sciences, Showa University, Hatanodai 1-5-8, Shinagawa, Tokyo 142-8555, Japan; kfukuchi@med.showa-u.ac.jp; 2Department of Microbiology, St. Marianna University School of Medicine, Sugao 2-16-1, Miyamae, Kawasaki 216-8511, Japan; takuro.koshikawa@marianna-u.ac.jp (T.K.); asai@ac.shoyaku.ac.jp (D.A.); takeh@marianna-u.ac.jp (H.T.); 3Laboratory of Microbiology, Showa Pharmaceutical University, 3-3165 Higashi-Tamagawagakuen, Machida, Tokyo 194-8543, Japan; kanamoto@ac.shoyaku.ac.jp; 4Division of Microbiology, Meikai University School of Dentistry, Keyakidai 1-1, Sakado, Saitama 350-0283, Japan; inomata@dent.meikai.ac.jp; 5Research Institute of Odontology (M-RIO), Meikai University, Keyakidai 1-1, Sakado, Saitama 350-0283, Japan; 6Research Laboratory, Research and Development Division, Nippon Paper Industries Co., Ltd., 5-21-1 Oji, Kita-ku, Tokyo 114-0002, Japan; hikaru-aimi@nipponpapergroup.com; 7Chemical Sales Department II, Chemical Sales Division, Nippon Paper Industries Co., Ltd., 4-6 Kandasurugadai, Chiyoda-ku, Tokyo 101-0062, Japan; y.kikkawa@nipponpapergroup.com

**Keywords:** lignosulfonates, anti-HIV, anti-HSV, quick inactivation of virus, water solubility, MTT assay, oral application

## Abstract

**Background:** Very few studies of the antiviral potential of lignosulfonates have been published. With the aim of oral application, among various groups of natural products, the relative antiviral potency of lignosulfonate and its ability to rapidly inactivate viruses were investigated. **Methods:** As target cells, MT-4 cells in suspension and attached Vero cells were used for infections with human immunodeficiency virus (HIV) and human herpes simplex type-1 virus (HSV). Mock- or virus-infected cells were incubated for 3–5 days with various concentrations of test samples, and the viable cell number was determined with the MTT method. For the shorter exposure experiments, higher titers of HIV or HSV were exposed to test samples for 10 or 3 min, diluted to a normal multiplicity of infection (MOI), and applied to the cells. Antiviral activity was quantified by using the chemotherapy index. **Results:** In the long-exposure system, lignosulfonates showed comparable anti-HIV activity with those of AZT, ddC, and sulfated polysaccharides, and it exceeded those of hundreds of tannins and flavonoids. When the exposure time was shortened, the chemotherapeutic index of the lignosulfonates for HIV was increased 27-fold. At a physiological pH, lignosulfonate showed higher anti-HIV activity than commercial alkali-lignin, dealkali-lignin, and humic acid, possibly due to the higher solubility and purity. **Conclusions:** With their rapid virus-inactivation capabilities, lignosulfonates may be useful for the prevention or treatment of virally induced oral diseases.

## 1. Introduction

Due to the worldwide spread of COVID-19, the antiviral potential of natural products has been re-evaluated [1,2]. This has been reflected by the increase in the number of papers on the antiviral activity of natural products (plant extracts, polysaccharides, flavonoids, tannins, and lignins) over the 30 years from 1991 to 2020 (Figure 1). The number of publications on Traditional Chinese Medicine (TCM) rapidly increased in 2020 and surpassed those on plant extracts, which were followed by flavonoids, while publications on sulfated polysaccharide and lignin stayed unchanged or increased very slowly (Figure 1).

Publications on subgroups (Groups 1–5, Table 1) were surveyed. The number of published papers on anti-HIV (human immunodeficiency virus) and anti-HSV (herpes simplex virus type-1) activity was approximately 10% of the number of publications on all viruses. With respect to plant extracts (Group 1), there were more than 25 times more publications on Traditional Chinese Medicine (TCM) than on Traditional Japanese Medicine (Kampo Medicine), hot-water extracts, and alkaline extracts. For flavonoids (Group 2), antiviral studies on flavonol, isoflavone, and commercially available resveratrol and curcumin were the most abundant. Among tannin-related compounds, gallic acid (a component unit of tannin), epigallocatechin gallate, and catechin were the most popular. However, the antiviral activity of most of these lower-molecular-weight compounds was low [3]. For polysaccharides (Group 2), the number of publications on sulfated polysaccharides represented approximately 10% of the publications on all polysaccharides. On the other hand, the numbers of papers on the antiviral activity of natural and synthetic lignin and lignosulfonate were much lower. When a search was conducted for the antiviral activity of lignosulfonate, only six papers were found: four papers dealt with anti-HIV and/or anti-HSV activity [4,5,6,7], one paper with anti-diabetic activity [8], and one paper with anti-inflammatory activity [9]. The anti-HIV mechanism of lignosulfonate is the inhibition of viral penetration via the interaction with viral envelop glycoprotein [5,6], which is similar to the mechanism of sulfated polysaccharide [10,11]. However, the potency of lignosulfonate relative to that of other natural products has not yet been reported. 

Lignins, the major class of natural products present in the natural kingdom, are formed by the dehydrogenative polymerization of three monolignols: *p*-coumaryl, *p*-coniferyl, and sinapyl alcohols [12]. Some polysaccharides in the cell walls of lignified plants are linked to lignin, forming lignin-carbohydrate complexes (LCCs). The carbohydrate portion of LCC is composed of heterologous sugars such as glucose, arabinose, mannose, galactose, fucose, and occasionally uronic acids, depending on the plant species. When the ratio of polysaccharides to phenylpropanoids varies, the heterogeneity in the acidity, water solubility, ethanol insolubility, and molecular weight are generated. The molecular weight of LCCs ranges from 2.0 to 35 kDa [13,14,15]. Due to the large molecular mass, the biological availability of lignin was disappointingly low. Only a small % of orally administered ^125^I-lignin was recovered from the blood, while most of the input radioactivity was excreted into feces [16]. On the other hand, lignin showed a strong affinity to the influenza virus (demonstrated by sucrose gradient centrifugation) and instantly eliminated the lethality of influenza virus infection against monkeys [17]. These data suggest that the antiviral activity of lignosulfonate may be maximized by its direct application to the affected part of the body, such as the oral cavity, where numerous bacteria and viruses are present.

The importance of oral health has been reported by many papers. A population-based longitudinal study demonstrated the association of tooth loss rate with the risk of mild cognitive impairment in older adults [18]. Oral function is essential for nutrient intake and can be restored by dental prosthetic treatments in patients with tooth loss [19]. Supportive periodontal treatment with repeated oral hygiene education increased the cumulative survival rates [20]. Many viruses such as human immunodeficiency virus (HIV), herpes simplex virus (HSV) 1 and 2, human cytomegalovirus, Epstein-Barr virus, BK virus, JC virus, and adeno-associated virus (AAV) stimulate the development of genital- and oral-HPV-associated carcinomas [21], and possibly periodontal inflammation [22]. Epigallocatechin-3-gallate (EGCG) gel [23,24] has been applied as a dental anti-caries, although the anti-HIV activity of condensed tannins, including EGCG, was very low [25].

We have previously reported that (i) the alkaline extracts of several tea leaves [26] and licorice root [27] shows higher anti-HIV activity than the corresponding hot water extracts; (ii) hot water extracts such as 10 Kampo medicines and their 25 constituent plant extracts [28], and three Chinese herbal extracts (TCM) from *Drynaria baronii*, *Angelica sinensis,* and *Cornus officinalis.* Sieb. et Zucc [29] showed little or no anti-HIV activity; (iii) purification of lignin from the alkaline extract of the leaves of Sasa sp. by repeated acid-precipitation and solubilization steps resulted in the increase in anti-HIV activity [30]; (iv) lignin sulfonate showed much higher water solubility than lyophilized lignin powder (as shown by this study). In the present pilot experiments, the anti-HIV activity of six newly manufactured lignosulfonates was first investigated in comparison with those of other groups of natural products [3] and commercially available positive controls such as reverse transcriptase inhibitors [azidothymidine (AZT), 2′,3′-dideoxycytidine (ddC)], and sulfated polysaccharides (dextran sulfate, curdlan sulfate). Since alkaline extract of *Sasa* sp. (SE) that contains LCC [31] has been reported to rapidly inactivate both HIV and HSV [32], whether lignosulfonates rapidly inactivate both HIV and HSV was next investigated for future application to dentistry. Our goal is to manufacture the lignosulfonate-containing gargle or mouth wash that should instantly inactivate HSV and HIV.

## 2. Materials and Methods

### 2.1. Materials

The following chemicals and reagents were obtained from the indicated companies: Eagle’s minimum essential medium (MEM) (Gibco BRL, Grand Island, NY, USA); fetal bovine serum (FBS), 3-(4,5-dimethylthiazol-2-yl)-2,5-diphenyltetrazolium bromide (MTT), resveratrol, azidothymidine (AZT), 2′,3′-dideoxycytidine (ddC) (Sigma-Aldrich Inc., St. Louis, MO, USA); dimethyl sulfoxide, dextran sulfate (DS) (5 kDa) (Wako Pure Chemical Ind., Ltd., Osaka, Japan); curdlan sulfate (79 kDa) (Ajinomoto Co., Inc., Tokyo, Japan); and PVP-I (Showa Seiyaku Co. Ltd., Tokyo, Japan); alkali-lignin, dealkali-lignin, humic acid, sodium lignin sulfonate (Tokyo Chemical Industry Co., Ltd., Tokyo, Japan). Culture plastic dishes and plates (96-well) were purchased from Becton Dickinson Labware (Franklin Lakes, NJ, USA).

### 2.2. Preparation of Lignosulfonate

All lignosulfonates samples used in this study were provided by Nippon Paper Industries Co., Ltd., Tokyo, Japan. Their preparative method, molecular weight (determined by gel permeation chromatography), and purity are shown in Figure 2. Lignosulfonate A is derived from sulfite spent liquor and contains sodium lignosulfonate as the main component. Lignosulfonate B and Lignosulfonate C are purified high-molecular-weight sodium lignosulfonates, and the latter was subjected to additional modifications, including oxidation. Lignosulfonate D is a purified and partly desulfonated sodium lignosulfonate produced under alkaline conditions at a high temperature and pressure. Lignosulfonate E is a type similar to Lignosulfonate D but has higher purity due to the inclusion of a washing procedure. Lignosulfonate F is also a similar type as Lignosulfonate D. Lignosulfonate F is weakly acidic, while Lignosulfonate D is weakly alkaline, and the water solubility of Lignosulfonate F is lower than that of Lignosulfonate D. Lignin, including lignosulfonate, and does not have uniform structures since lignin is formed by polymerization of primary lignin precursors in a relatively random way called ‘Dehydrogenative polymerization.’ Thus, only tentative structures of lignosulfonates have been proposed [33].

### 2.3. Assay for Antiviral Activity

#### 2.3.1. Assay for Anti-HIV Activity (Using Both Long- and Short-Exposure Systems)

Sample was dissolved at 10 mg/mL with phosphate-buffered saline (PBS, pH 7.4) or 1.39% NaHCO_3_, vortexed, and shaken overnight at 4 °C. After centrifugation, the supernatant was collected and sterilized by passing through the 0.45 µm membrane filter. For the long-exposure protocol, MT-4 cells (3 × 10^4^ cells) were incubated with HIV at multiplicity of infection (MOI) of 0.01. HIV-and mock-infected MT-4 cells were cultured for 5 days with serial dilutions of lignosulfonates, and the cell viability was quantified with the MTT method [34] (Figure 3A). For the short-exposure protocol, the 20-fold concentrated HIV (MOI = 0.2) was incubated with lignosulfonates at different concentrations for 10 min, then the mixtures were diluted 20-fold and then added to MT-4 cells, and cultured for 5 days at the MOI of 0.01. For the evaluation of cytotoxicity, MT-4 cells were incubated with lignosulfonates at the indicated concentrations for 10 min. After removal of lignosulfonates by centrifugation, MT-4 cells were rinsed with PBS, and then cultured for 5 days to determine the viable cell number with the MTT method (Figure 3B).

From the dose-response curve, the 50% cytotoxic concentration (CC_50_) in mock-infected cells and the 50% effective concentration (EC_50_) in HIV-infected cells were determined. The EC_50_ value was defined as the concentration at which the viability was restored to 50% that of the mock-infected cells. The anti-HIV activity was evaluated by the selectivity index (SI), which was calculated using the following equation: SI = CC_50_/EC_50_ (Figure 3C).

#### 2.3.2. Assay for Anti-HSV Activity (Using Only the Short-Exposure System)

Sample was dissolved at 10 mg/mL with phosphate-buffered saline (PBS, pH 7.4) or 1.39% NaHCO_3_, vortexed, and shaken overnight at 4 °C. After centrifugation, the supernatant was collected and then sterilized by passing through 0.45 µm membrane filter. Mock-infected Vero cells, derived from African green monkey kidney cells, were first treated for short period (3 min) with various concentrations of test samples without HSV, then the supernatant that contained sample was removed by suction. The cells were washed once with PBS and incubated for 3 days in the fresh culture medium. The viability of mock-infected cells was determined by the MTT method. The CC_50_ was determined from the dose-response curve [32].

For the virus infection, 100-fold concentrated HSV (MOI = 1) was mixed with the test samples and stood for 3 min to expose the cells to extremely higher concentrations of samples. Then, virus titer was reduced 100-fold up to the MOI of 0.01, added to the cells, and incubated for three days to determine the EC_50_ (Figure 4A). The HSV infection at MOI of 0.01 reduced the cell viability to 20~40% (Appendix A). If higher titer of MOI was used, the complete recovery of the cell viability by any sample could not be guaranteed. Three days’ incubation was the optimal condition for the quantitative determination of anti-HSV activity.

From the dose-response curve, the CC_50_ value in mock-infected cells and the EC_50_ value in HSV-infected cells were determined. EC_50_-I was defined as the concentration that restored the viability to the midpoint between that of HSV-infected cells and that of mock-infected cells. EC_50_-II was defined as the concentration that restored the viability to 50% that of mock-infected cells. Anti-HSV activity was evaluated by the selectivity index (SI-I and SI-II), determined using the following equation: SI-I = CC_50_/EC_50_-I; SI-II = CC_50_/EC_50_-II (Figure 4B) [32].

### 2.4. Statistical Treatment

Experiment values are expressed as the mean ± S.D. of triplicate unless otherwise stated.

## 3. Results

### 3.1. Lignosulfonate Rapidly Inactivates HIV

#### 3.1.1. Lignosulfonate Is the First Class of Anti-HIV Agent (Long-Exposure Experiment)

MT-4, HTLV-I-transformed T-cell line was used to measure the anti-HIV activity of samples. HIV infection (M.O.I = 0.01) induced the complete loss of viability, accompanied by syncytia formation and the inhibition of HIV-specific antigen expression [7]. The addition of lignosulfonate B rescued the cells from the cytopathic effect of HIV infection. Lignosulfonate B showed potent anti-HIV activity (SI > 1210), comparable with those of sulfated polysaccharides (dextran sulfate, curdlan sulfate) and reverse transferase inhibitors (azidothymidine (AZT) and 2′,3′-dideoxycytidine (ddC)) (Figure 5). Repeated experiments revealed that lignosulfonate B, dextran sulfate, and curdlan sulfate exclusively eliminated the HIV-induced cytopathic effect, whereas the protective effects of AZT and ddC were not complete (Figure 5 and Appendix A).

Next, the anti-HIV activity of these samples was compared with those of other natural products (Figure 6). Lignosulfonate B (B), sulfated polysaccharides (sodium paramylon sulfate [35], dextran sulfate, and curdlan sulfate) (E), reverse transcriptase inhibitors (AZT, ddC) (A), indicated by orange color, showed the highest anti-HIV activity. This was followed by natural and synthetic lignin (dehydrogenation polymers of phenylpropanoids) (C, D) [36], and alkaline plant extract (G), indicated by green color.

On the other hand, the anti-HIV activity of polysaccharides (unsubstituted or substituted with *N*,*N*-dimethylaminoethyl, *N*,*N*-diethylaminoethyl, 2-hydroxy-3-trimethylammoniopropyl, and carboxymethyl group) (F) [35], hot water plant extract (Kampo medicines and its constituent plant extracts) (H) [28], tannins (I) [25], and flavonoids and chromones (J) (indicated by blue bars) [3,37,38,39] were much less. It should be noted that the anti-HIV activity of hydrolyzable tannins increased with oligomerization (monomer < dimer < trimer < tetramer) [25].

#### 3.1.2. Enhancement of Anti-HIV Potential of Lignosulfonate by Shortening Treatment Time

We next investigated the possibility of rapid inactivation of HIV-1 by the short exposure to lignosulfonates (A, B, C, D, E, and F). The 20-fold concentrated HIV (MOI = 0.2) was first exposed to higher concentrations of samples for 10 min, diluted, and then added to MT-4 cells. The viable cell number was determined after 5 days with the MTT method. Aliquots of cells were treated with higher concentrations of test samples, washed with PBS, and incubated for 5 days to determine the CC_50_ value. The SI value was determined by the following equation: SI = CC_50_/EC_50_ (Figure 7). Lignosulfonate B showed the highest SI value (SI = 32854), followed by lignosulfonate C (SI > 17982) > lignosulfonate E (SI > 16529) > lignosulfonate D (SI > 15652) > lignosulfonate F (SI > 9733) > lignosulfonate A (SI > 261). Slight but apparent anti-HIV activity of lignosulfonates B, C, E, and F was detected at the concentration of as little as 5.12 ng/mL.

By shortening the exposure time, anti-HIV activity of lignosulfonate B was increased by 27.2-fold, based on the specific index (from SI = 1210 to SI = 32854), or 25.7-fold, based on the EC_50_ value (=0.826/0.0322), (Table 2). This indicates that more pronounced anti-HIV potential of lignosulfonate would be expected when it is used as quick inactivator of a virus. On the other hand, the increase in anti-HIV-activity of dextran sulfate, curdlan sulfate, AZT and ddC by shortening the exposure time was only 0.7-fold {= [(0.0395 + 0.0593 + 0.0419)/3]/0.0645}, 2.6-fold {= [(0.668 + 0.208 + 0.133)/3]/0.13}, 0.9-fold {= [(0.0168 + 0.0165 + 0.0098)/3]/0.0168}, and 2.8-fold {= [(3.07 + 3.62 + 3.88)/3]/1.25}, respectively (Table 2 and Appendix A).

We found that lignosulfonates B and C with higher molecular weights showed higher anti-HIV activity than other lignosulfonates. Lignosulfonate A, which contains 50% impurities (Figure 2), showed the lowest anti-HIV activity (SI > 261) (Table 2).

Since the anti-HIV activity (assessed as SI value (=CC_50_/EC_50_) of four positive controls (dextran sulfate, curdlan sulfate, AZT, ddC) changed considerably from experiment to experiment (Appendix A), the anti-HIV activity of lignosulfonate samples and these positive controls were assayed at the same time. The dose-response curve demonstrated that most of the CC_50_ values of lignosulfonates scaled over the maximum value due to low cytotoxicity, making the EC_50_ value dominant for SI determination. Variation of standard deviation (SD) of the EC_50_ value in each point was at most 5~10%, in contrast to the great difference of SI values between samples. For example, the SI value of lignosulfonate A was approximately100-fold lower than other lignosulfonates B, C, D, E and F (that showed comparable or much higher SI values than positive controls). This indicates the obtained SI values are reliable. 

### 3.2. Lignosulfonate Rapidly Inactivates HSV (Short Exposure)

The possibility of rapid inactivation of HSV-1 by short exposure to lignosulfonates A, B, C, D, E, and F was next investigated. One hundred-fold concentrated HSV was exposed for 3 min to various concentrations of test samples. The virus-containing medium was diluted 100-fold and added to the Vero cells. After incubation for 3 days, the EC_50_ value was determined. Since HSV infection could not completely eradicate the viable cells (leaving 23.4–48.2% of viable cells), two sets of EC_50_ values, that is, EC_50_-I and EC_50_-II, were calculated as described in Materials and Methods. The CC_50_ value was determined by incubating Vero cells with different concentrations of test samples, washing with PBS, and further incubating for 3 days (Figure 5). Accordingly, two SI values were produced: SI-I (=CC_50_/EC_50_-I) and SI-II (=CC_50_/EC_50_-II). As a control, commercially available alkali-lignin, dealkali-lignin, and humic acid (chromogenic and structurally irregular organic assemblies that are widespread in soils, rivers, oceans, and coal-related natural resources) [40] were used.

Among six lignosulfonates, lignosulfonate B showed the highest anti-HSV activity based on the SI value. On the other hand, lignosulfonate A and F showed the lowest SI value, regardless of dissolving them with an alkaline solution (1.39%NaHCO_3_, pH 8.0) or with a neutral buffer (phosphate-buffered saline, pH 7.4) (Figure 8, Table 3). Highly purified lignosulfonate E (purity: 97%) showed higher anti-HSV activity when it was dissolved in PBS (−).

Alkali-lignin, dealkali-lignin, and humic acid showed anti-HSV activity comparable with that of lignosulfonate B; when they were dissolved with an alkaline solution, however, their anti-HIV activity was dramatically reduced by dissolving them with neutral buffer (Figure 9, Table 3). This indicates that solubility is an essential factor that determines the anti-HSV activity. 

## 4. Discussion

The present study further confirmed the excellent antiviral activity of lignosulfonate [4,5,6,7], and its antiviral activity was comparable with those of sulfated polysaccharides and much higher than those of other lower polyphenols (Figure 6). During the isolation process, lignin forms the complex with polysaccharides. Anti-influenza A virus and HIV activities of lignin-carbohydrate complex (LCC) were considerably reduced by treatment with sodium chlorite but were not affected by sulfuric acid or trifluoroacetic acid [7,41,42], confirming that the active principle of LCC is the polymerized phenolic structure [29], but not carbohydrate. This was supported by the excellent antiviral activity of synthetic lignin (dehydrogenation polymer of phenylpropanoids) without polysaccharides [36]. Both lignin and polysaccharide provide scaffolds for the attachment of a sulfonate or sulfate group. The antiviral activity of lignosulfonate may depend on the molecular size since lignosulfonates B and C, which have higher molecular weight than other lignosulfonates, showed higher anti-HIV activity (Table 2). The substitution ratio of the sulfate group may also affect the antiviral activity of sulfated polysaccharides [35].

The anti-HSV activity of commercially available lignosulfonate, alkali-lignin, dealkali-lignin, and humic acid depends on their solubility. When these substances were dissolved with 1.39% NaHCO_3_ (pH 8.0), they showed anti-HSV activity comparable with each other (Figure 8). However, when they were dissolved with a neutral buffer, PBS (pH 7.4), their anti-HSV activity diminished (Figure 9), in contrast to lignosulfonates, which dissolved well in both alkaline and neutral solutions (Figure 8).

The purity of the sample is another factor that determines the antiviral activity. Lignosulfonate A (purity: 50%) showed much lower anti-HIV activity (SI > 261) than lignosulfonate B-F (purity: 92, 81, 87, 97 and 89, respectively) (SI >9733–32854) (Figure 7). Commercial lignosulfonate showed slightly lower anti-HSV activity (SI >3.7, >4.5) when dissolved in PBS (as shown in Appendix A), as compared with those of lignosulfonate B (SI = 14.1, >21.7) and lignosulfonate E (SI = >16.1, >21.7). 

The anti-HIV activity of lignosulfonate B was enhanced 27.2-fold when the treatment period was shortened (Table 2). On the other hand, the reduction in the exposure time did not enhance the anti-HIV activity of sulfated polysaccharides (dextran sulfate, curdlan sulfate) and reverse transcriptase inhibitors (AZT, ddC). This may be explained by the fact that lignosulfonate B, with both a polymerized phenolic structure and sulfonate groups, may adsorb the virus more quickly than sulfated polysaccharides and reverse transcriptase inhibitors. Although the anti-HIV activity of resveratrol and curcumin is two or three orders lower than lignosulfonate (Figure 6), their antiviral activity may be enhanced by adopting a short treatment schedule. 

Alkaline extract of *Sasa* sp. (SE) showed comparable anti-HIV activity with that of natural and synthetic lignin [36], leading to the manufacture of toothpaste that includes 29% SE [43]. The present study demonstrated that lignosulfonate B showed 28–42-fold higher anti-HIV activity (EC_50_ = 0.0322 μg/mL) (Table 2) than SE (EC_50_ = 0.89–1.35 μg/mL) [32], and rapidly inactivated HIV and HSV. Considering that the chemotherapy index of SE is one order higher than that of povidone iodine (the most popular gargling solution in Japan) (SI = 26.1–31.6 vs. 2–3.1) [32], the chemotherapy index of lignosulfonate is estimated to be two or three orders higher than that of povidone iodine. The rapid virus inactivation activity of lignosulfonate suggests its applicability as a toothpaste and mouth washer to prevent viral infection. 

Lignosulfonate B, but not AZT and ddC, completely inhibited the cytopathic effect of HIV infection (Figure 7). Considering the synergism of SE and antiviral agents [44], the combined treatment of lignosulfonate and AZT or ddC, having different cellular targets on HIV-induced cytopathy, may be efficacious. 

Oral health is important for living a longer life since the oral cavity contains a lot of microorganisms such as bacteria and viruses. Most researchers have focused on bacteria affecting quality of life. However, not only bacteria but also viruses may have ill effects on oral function. Oral herpes virus is very common and often causes debilitating infectious diseases in patients, affecting oral health and having serious psychological implications [45]. HIV is a blood-borne virus that can enter the mouth via gingival crevicular fluid and hides in the oral HIV reservoir [46]. Due to the behavioral changes in sexual conduct, conventional oral transmission of sexually transmitted HIV infections has been progressively recognized. Pediatric infections could also be the result of breastfeeding by the oral route. Pathogenic viruses such as HIV and HSV come with an array of complexities in the oral cavity, and therefore, keeping a clean mouth is essential for living longer with a high QOL. Oral intake of SE significantly improved the symptoms of oral lichenoid dysplasia (as evidenced by the reduced area of white steaks and the decline of the salivary concentrations of interleukin-6 and -8) [47], possibly by its potent anti-inflammatory activity [48].

SARS-CoV-2 exhibits several complications, such as lung damage, blood clot formation, respiratory illness, organ failures [49], and neurologic manifestations such as taste [50] and smell loss [51] in most patients. At present, there is no report available for anticoronavirus activity of both lignosulfonate and sulfated polysaccharides. Since these substances prevent virus entry into host cells, they can be used as candidate drugs, adjuvants in vaccines, or in combination with other antivirals, antioxidants, and immune-activating nutritional supplements and antiviral materials to prevent SARS-CoV-2 infection [49]. The anti-diabetic [8], anti-inflammatory [9], and antiviral activity (this study) of lignosulfonate, and the neuroprotective activity of lignin [52] and SE [53] may be advantageous for the treatment of COVID-19. 

Lignin that contains mannose in the polysaccharide portion has been reported to stimulate the expression of the mRNA of dectin-2 [54], a C-type lectin receptor that recognizes high-mannose polysaccharides [55]. Research into the distribution of dectin-2 in oral tissues and its role in immunopotentiation is underway. 

## 5. Conclusions

Lignin is a main constituent of wood, such as cellulose and hemicellulose, and is found abundantly in nature. The present study demonstrates for the first time that lignosulfonate is a first-class anti-HIV agent, and it rapidly inactivates HIV and HSV. These data suggest its possible utilization as an ingredient in gargle mouthwash and toothpaste. 

## Figures and Tables

**Figure 1 medicines-08-00056-f001:**
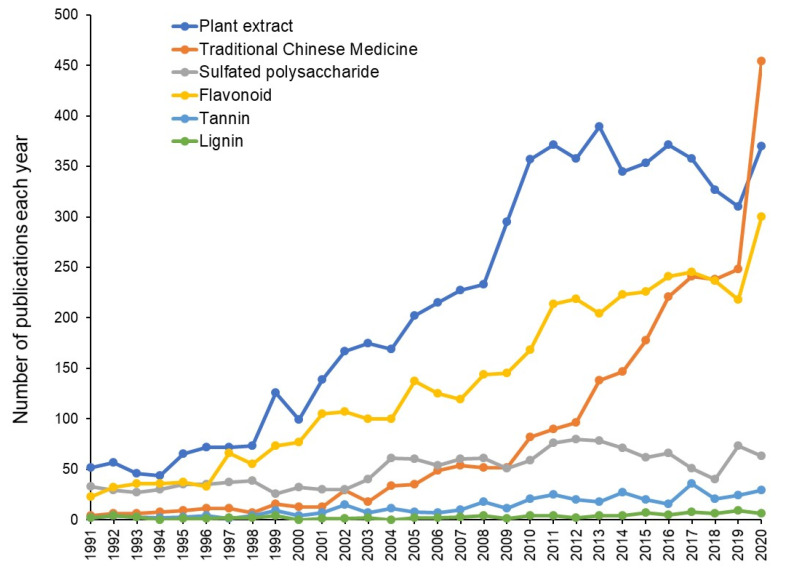
Number of publications on antiviral research of natural products over the 30 years from 1991 to 2020. Data were taken from Pubmed (14 April 2021).

**Figure 2 medicines-08-00056-f002:**
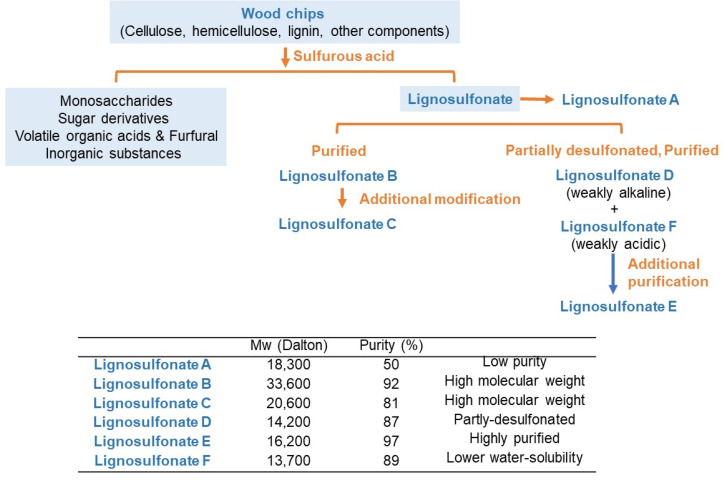
Preparative scheme of lignosulfonate. Purity and molecular weight were characterized by Nippon Paper Industries Co., Ltd., Tokyo, Japan.

**Figure 3 medicines-08-00056-f003:**
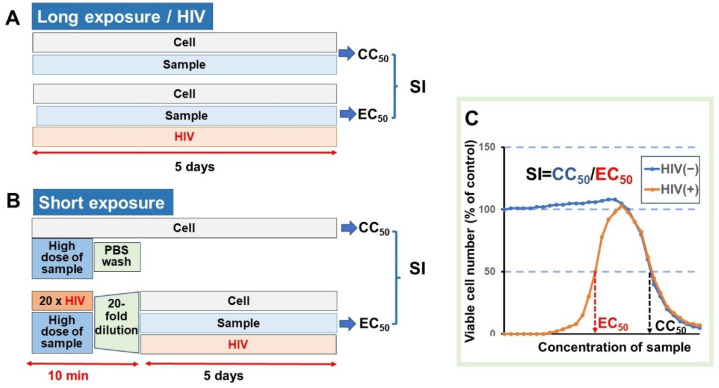
Experimental protocol for the determination of anti-HIV activity using long- (**A**) and short- (**B**) exposure assay systems, and the dose-response curve for the calculation of CC_50_, EC_50_, and SI values (**C**).

**Figure 4 medicines-08-00056-f004:**
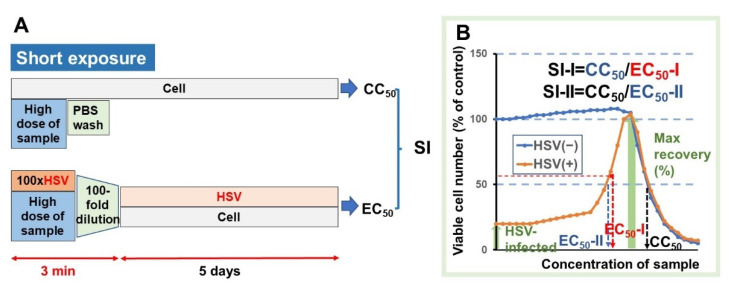
Experimental protocol for the determination of anti-HSV activity using the short-exposure assay system (**A**), and the dose-response curve for the calculation of CC_50_, EC_50_, and SI values (**B**).

**Figure 5 medicines-08-00056-f005:**
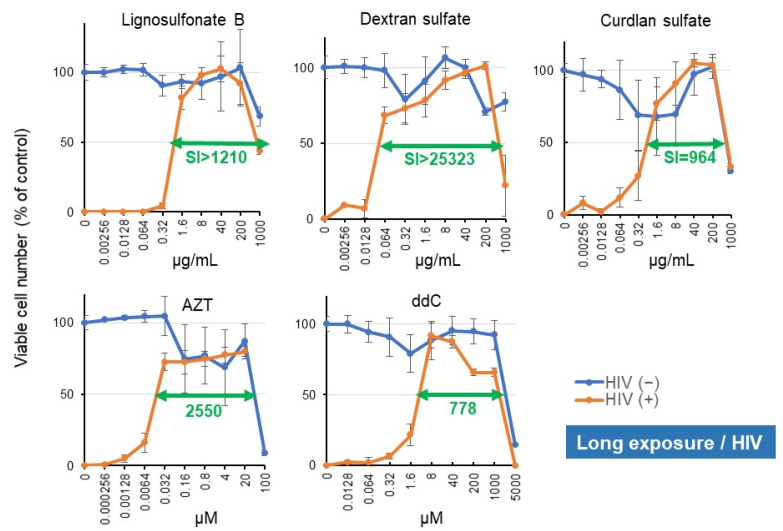
Lignosulfonate B showed comparable anti-HIV activity with sulfated polysaccharides and reverse transcriptase inhibitors. Mock- or HIV (M.O.I = 0.01)-infected MT-4 cells were incubated for 5 days with the indicated concentrations of either lignosulfonate B or four positive controls (dextran sulfate, curdlan sulfate, AZT, ddC), and then the viable cell number was determined by the MTT method, as described in Figure 3A. Each value represents the mean ± S.D. of triplicate assays.

**Figure 6 medicines-08-00056-f006:**
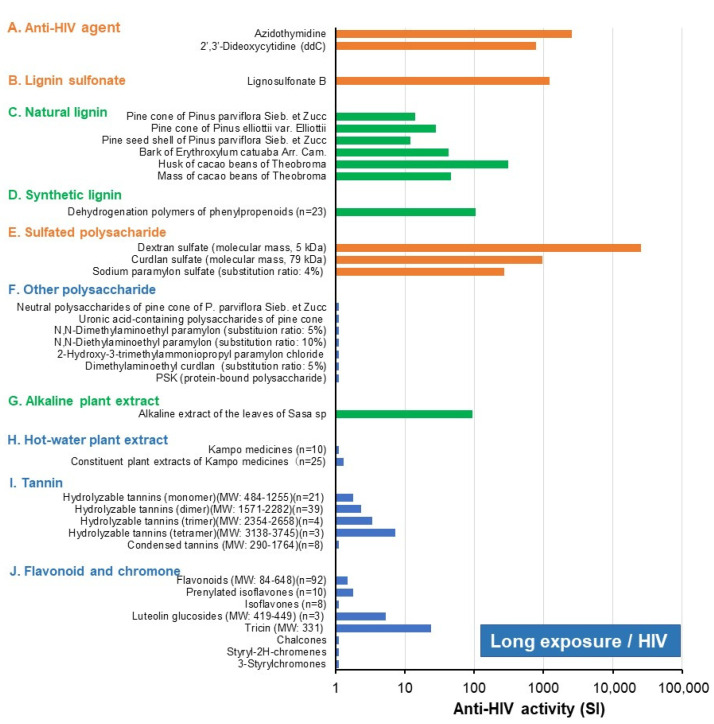
Lignosulfonate B and sulfated polysaccharides are potent anti-HIV substances. The data of lignosulfonate B (B) and anti-HIV agents (dextran sulfate, curdlan sulfate, AZT, and ddC) (indicated by orange color) were taken from the present study (Figure 2). Data of synthetic lignin were taken from [36], paramylon derivatives from [35], Kampo medicine from [28], hydrolyzable and condensed tannins from [25], chromones and chalcones (J) from [37,38,39], and other data from [3].

**Figure 7 medicines-08-00056-f007:**
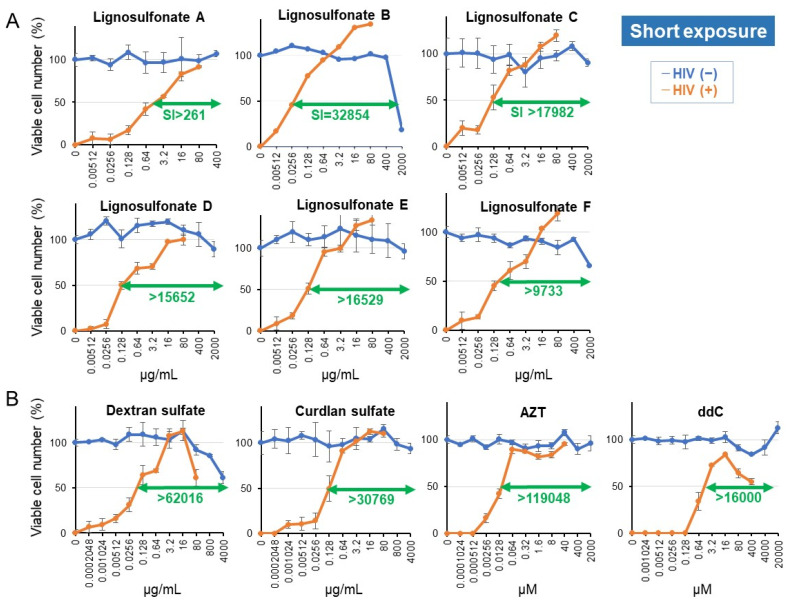
Anti-HIV activity of 6 lignosulfonates (**A**) and positive control (**B**), evaluated by the short-exposure protocol. HIV was incubated with the indicated concentrations of lignosulfonates for 10 min, then the mixtures were added to MT-4 cells and cultured for 5 days. For the evaluation of cytotoxicity, MT-4 cells were incubated with the indicated concentrations of lignosulfonates for 10 min. After removing the lignosulfonates by centrifugation, MT-4 cells were cultured for 5 days. Viable cell number was determined with the MTT method. Each value represents the mean ± S.D. of triplicate assays. It should be noted that HIV infection (M.O.I. = 0.01) induced the complete loss of cell survival, which was returned to control level by the addition of lignosulfonate.

**Figure 8 medicines-08-00056-f008:**
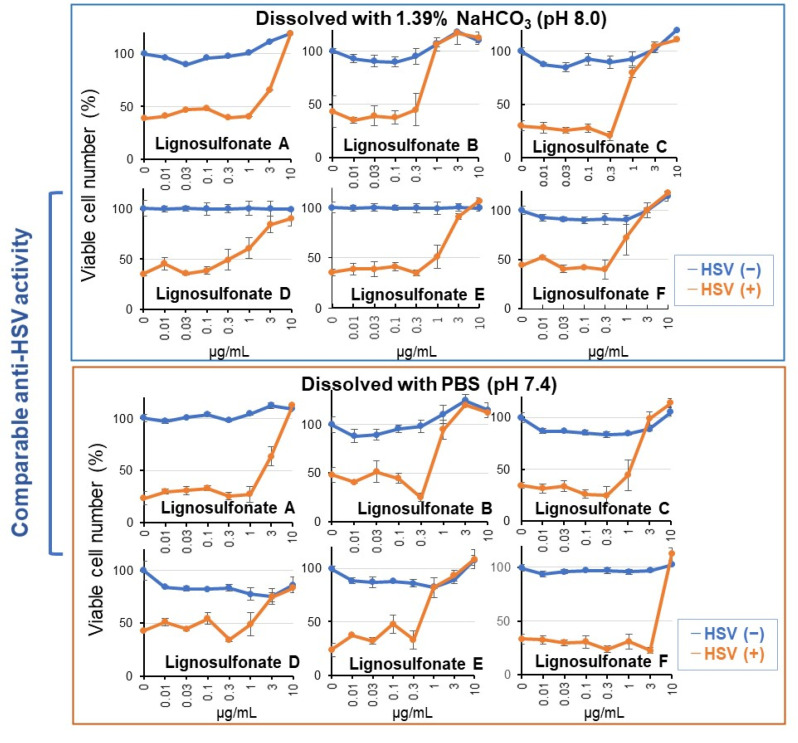
Lignosulfonates showed potent anti-HSV activity regardless of the pH of dissolving solution: 1.39% NaHCO_3_ (pH 8.0) or phosphate-buffered saline (PBS) (pH 7.4). It should be noted that the viability of HSV-infected cells was reduced to 23.4~48.2%.

**Figure 9 medicines-08-00056-f009:**
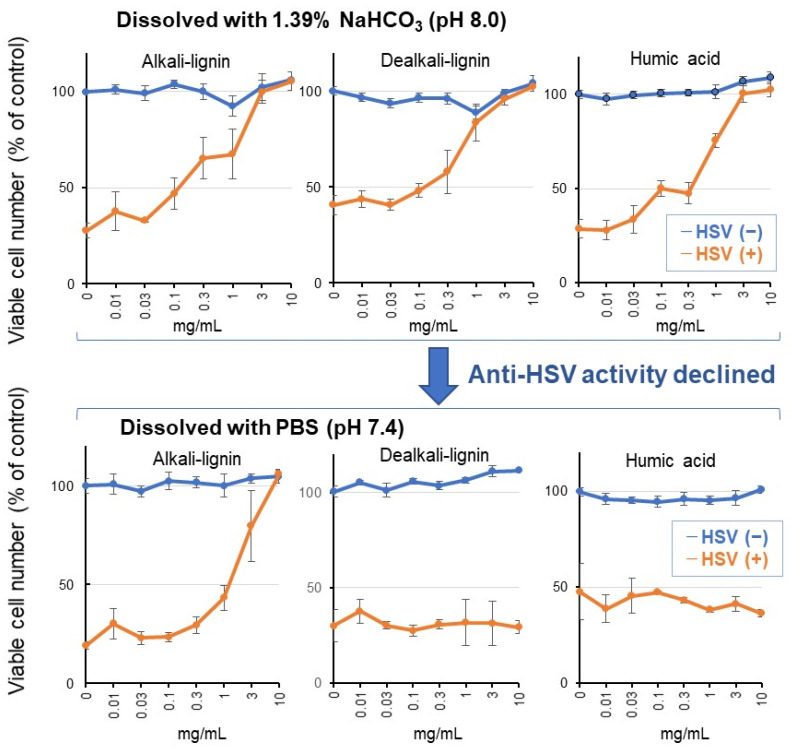
Anti-HSV activity of alkali-lignin, dealkali-lignin, and fumic acid declined when they were dissolved with PBS. Each value represents the mean ± S.D. of triplicate assays.

**Table 1 medicines-08-00056-t001:** Number of papers of antiviral research in each classified group (Pubmed search, 14 April 2021).

		Number of Papers Reported Thus Far		
Classification	Subgroup	Antiviral	Anti-HIV	Anti-HSV
Plant extracts	Plant extracts	6173	564	126
(Group 1)	Traditional Chinese medicine	2429	119	20
	Kampo medicine	67	3	2
	Hot water extract	79	9	6
	Alkaline extract	92	16	4
Flavonoid	Flavonoid	3911	234	48
(Group 2)	Flavonol	596	19	5
	Flavone	4	0	0
	Flavanone	249	18	1
	Isoflavone	513	12	3
	Pterocarpan	12	3	0
	Coumestan	5	1	0
	Resveratrol	328	25	2
	Curcumin	488	31	0
	Tricin	15	2	0
Tannins	Tannin	388	60	16
(Group 3)	Hydrolyzable tannin	124	11	7
	Condensed tannin	99	10	2
	Procyanidine	51	8	2
	Gallic acid	322	28	18
	Epigallocatechin gallate	260	21	1
	Catechin	487	38	8
Polysaccharides	Polysaccharide	16165	656	51
(Group 4)	Sulfated polysaccharide	1588	229	31
Lignin	Lignin	98	32	5
(Group 5)	Synthetic lignin	10	6	0
	Lignosulfonate	5	3	1
	*p*-Coumaric acid	28	8	1

**Table 2 medicines-08-00056-t002:** Short exposure of lignosulfonate induced one order higher anti-HIV activity than long exposure.

Test Compound	Data Source	CC_50_ (μg/mL)	EC_50_ (μg/mL)	SI
<Short exposure>				
Lignosulfonate A	Figure 4	>400	1.53	>261
Lignosulfonate B		1056	0.0322	32,854
Lignosulfonate C		>2000	0.111	>17,982
Lignosulfonate D		>2000	0.128	>15,625
Lignosulfonate E		>2000	0.121	>16,529
Lignosulfonate F		>2000	0.205	>9733
Dextran sulfate		>4000	0.0645	>62,016
Curdlan sulfate		>4000	0.130	>30,769
AZT (μM)		>2000	0.0168	>119,048
ddC (μM)		>20,000	1.25	>16,000
<Long exposure>				
Lignosulfonate B	Figure 2	>1000	0.826	>1210
Dextran sulfate		>1000	0.0395	>25,323
Curdlan sulfate		644	0.668	964
AZT (μM)		42.9	0.0168	2550
ddC (μM)		2387	3.07	778
Dextran sulfate	Appendix A	138	0.0593	2321
Curdlan sulfate	(Exp. 1)	914	0.208	4392
AZT (μM)		167	0.0165	10,090
ddC (μM)		2450	3.62	677
Dextran sulfate	(Exp. 2)	155	0.0419	3703
Curdlan sulfate		480	0.133	3614
AZT (μM)		47.4	0.00980	4838
ddC (μM)		>5000	3.88	>1289

**Table 3 medicines-08-00056-t003:** Lignosulfonates showed higher anti-HSV activity than other lignified materials when they were dissolved with neutral buffer (PBS). We confirmed that repeated experiments produced similar results.

Test Compound	Solvent	Viability of HSV- Infected Cells (%)	CC_50_ (mg/mL)	EC_50_-I (mg/mL)	EC_50_-II (mg/mL)	SI-I	SI-II	Max Recovery (%)
<Lignosulfonate>								
Lignosulfonate A	NaHCO_3_	23.9	>10	3.5	1.6	>2.9	>6.3	119
	PBS	23.4	>10	2.8	2	>3.6	>5	112
Lignosulfonate B	NaHCO_3_	43.5	>10	0.52	0.35	>19.2	>28.6	117
	PBS	48.2	>10	0.71	0.46	>14.1	>21.7	120
Lignosulfonate C	NaHCO_3_	29.9	>10	0.74	0.55	>13.5	>18.2	110
	PBS	34.2	>10	1.6	1.2	>6.3	>8.3	114
Lignosulfonate D	NaHCO_3_	35.3	>10	1.4	0.32	>7.1	>31.3	91
	PBS	43	>10	2.6	1.1	>3.8	>9.1	83
Lignosulfonate E	NaHCO_3_	35.5	>10	1.6	0.96	>6.3	>10.4	106
	PBS	23.5	>10	0.62	0.46	>16.1	>21.7	108
Lignosulfonate F	NaHCO_3_	24	>10	1.1	0.45	>9.1	>22.2	117
	PBS	33	>10	5.5	4.4	>1.8	>2.3	112
<Other lignified materials>								
Alkali-lignin	NaHCO_3_	27.7	>10	0.28	0.12	>35.7	>83.3	100
	PBS	19.1	>10	1.8	1.4	>5.6	>7.1	106
Dealkali-lignin	NaHCO_3_	40.67	>10	0.56	0.13	>17.9	>76.9	102
	PBS	30.3	>10	-	-	-	-	-
Humic acid	NaHCO_3_	28.5	>10	0.6	0.34	>16.7	>29.4	100
	PBS	47.6	>10	-	-	-	-	-

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
