# Peer review of "Lignosulfonate Rapidly Inactivates Human Immunodeficiency and Herpes Simplex Viruses"

_medicines, 2021, doi:10.3390/medicines8100056_

Round 1

Reviewer 1 Report

This manuscript reported an interesting compound Lignosulfonate as a potential antiviral treatment. It is well written in English and the data is well presented. There are a few comments here:

  1. The logics behind Lignosulfonate usage is not clear, especially its antiviral potential. The authors listed multiple TCM but it is not clear how this linked with Lignosulfonate from the introduction.
  2. It seems the short exposure with high dose virus and Lignosulfonate showed better antiviral effects. However this is not really simulate the real infection. How could the authors apply this short exposure strategy to a real in vivo treatment scenario.
  3. For table 2 and table 3, there is lack of statistical analysis, such as how many experiment repeats, confidence interval of each measurement, how significant difference between different compounds.
  4. Figure 8, "reduced to 23.4 to 48.2%." It is confusing.

Author Response

Dear Reviewer 1:

Thank you for finding our report interesting and giving us very much constructive comments. We have responded to each comment as described below. I hope these corrections meet your criticisms. 

The logics behind Lignosulfonate usage is not clear, especially its antiviral potential. The authors listed multiple TCM but it is not clear how this linked with Lignosulfonate from the introduction.

Thank you for your suggestion of adding the reason why we select lignosulfonate. 

We have previously reported that (i) alkaline extract shows higher anti-HIV activity than hot water extract including 10 Kampo medicines and their 25 constituent plant extracts [26], and three Chinese herbal extracts from Drynaria baronii, Angelica sinensis and Cornus officinalis Sieb. et Zucc [27], (ii) the purification of lignin from alkaline extract by repeated acid-precipitation and solubilization steps, the specific activity of anti-HIV activity increased [28]), (iii) lignin sulfonate shows much higher solubility that lignin lyophilized after dialysis against water (as shown by this study).  This is the reason why we selected lignosulfonate.

These sentences were added in the introduction (line 101-106).

(Response)

It seems the short exposure with high dose virus and Lignosulfonate showed better antiviral effects. However this is not really simulate the real infection. How could the authors apply this short exposure strategy to a real in vivo treatment scenario.

(Response)

Our goal is to manufacture that lignosulfonate-containing gargle or mouth wash that should instantly inactivate HSV and HIV.

This statement was added in introduction (line 114-116)..

For table 2 and table 3, there is lack of statistical analysis, such as how many experiment repeats, confidence interval of each measurement, how significant difference between different compounds.

(Response)

For Table 2:

Since anti-HIV activity (assessed as SI value (=CC50/EC50) of four positive controls (dextran sulfate, curdlan sulfate, AZT, ddC) changed considerably from experiment to experiment (supplementary Table S1), anti-HIV activity of lignosulfonate samples and these positive controls were assayed at the same time.  Dose-response curve demonstrated that most of CC50 value of lignosulfonates scaled over the maximum value due to low cytotoxicity, making EC50 value dominant for SI determination.  Variation of standard deviation (SD) of EC50 in each point was at most 5~10%, in contrast to much difference of SI values between samples.  For example, the SI value of lignosulfonate A was approximate-ly100-fold lower than that of and others (lignosulfonate B, C, D, E, F) that showed comparable or much higher SI values than positive controls.  This indicates the obtained SI values are reliable. This statement is described in line 274-284.

As Table 3:

Due to high background level of viable cells after HSV-infection (mean value of 52 experiments = 20.1%, ranging from 5.2 to 50.0% (supplementary data of ref. 30), and variation of maximum recovery by addition of lignosulfonate, it is easy to variate around EC50 concentration.  When we find SD exceeded more than 20%, we have repeated the experiments, and find reproducible results.  We stated “We stated that “We confirmed that repeated experiments produced similar results” in Table 3.

Figure 8, "reduced to 23.4 to 48.2%." It is confusing.

(Response)

Thank you for pointing out our typographical errors.

We have corrected "reduced to 23.4 to 48.2%." to "reduced to 23.4 ~ 48.2%."

Other major corrections

Page 13, line 362:  “two or three fold higher than that of povidone iodine” corrected to “two or three order higher than that of povidone iodine”

Figure 6: E. “polyssacharide” was corrected to “polysaccharide”  

Figure 7  “Lignosufonate F” was corrected to “Lignosulfonate F”

Fourteen new references and one supplementary Table S1 were added.

Reviewer 2 Report

A well articulated manuscript.

Introduction could have more detail about lignins or any other anti-virals discussed currently used in dental products if any. The conclusion points out to how oral health is very important, but not much reference has been made in the introduction

You have mentioned different lignosulfonates with different purity. How was the purity recorded?

Are the different lignosulfonates chemically different? If so, a schematic or chemical structural representation would be useful.

What is the additional modification to convert lignosulfate B to lignosulfate C

Please describe the additional purification mention in conversion or lignosulfate F to E

Describe QOL (in line 328)

Author Response

Dear Reviewer 2:

Thank you for your evaluation of our manuscript and constructive comments.  We have responded to each comment as described below. I hope these corrections meet your criticisms. 

Introduction could have more detail about lignins or any other anti-virals discussed currently used in dental products if any. The conclusion points out to how oral health is very important, but not much reference has been made in the introduction

(Response)

We added the details about lignins, application of anti-viral catechin gels and how oral health is very important in the discussion, as follows.

Importance of oral health has been reported by many papers. A population-based longitudinal study demonstrated the association of tooth loss rate with the risk of mild cognitive impairment in older adults [18]. Oral function is essential for nutrient intake and can be restored using dental prosthetic treatments in patients with tooth loss [19]. Sup-portive periodontal treatment with repeated oral hygiene education increased the cumula-tive survival rates [20].  Many viruses such as human immunodeficiency virus (HIV), herpes simplex virus (HSV) 1 and 2, human cytomegalovirus, Epstein-Barr virus, BK virus, JC virus, and adeno-associated virus (AAV) stimulate the development of genital and oral HPV-associated carcinomas [21], and possibly periodontal inflammation [22].  Epigallo-catechin-3-gallate (EGCG) gel [23-24] has been applied in dental anti-caries, although condensed tannins including EGCG showed little or no anti-HIV activity [25]. (line 90-100)

You have mentioned different lignosulfonates with different purity. How was the purity recorded?

(Response)

We have changed the title of Figure 2 with addition of some information as described below.   

Figure 2. Preparative scheme of lignosulfonate. Purity and molecular weight were characterized by Nippon Paper Industries Co., Ltd.

Are the different lignosulfonates chemically different? If so, a schematic or chemical structural representation would be useful.

(Response)

Lignin, including lignosulfonate, does not have uniform structures because lignin is formed by polymerization of primary lignin precursors in a relatively random way called ‘Dehydrogenative polymerization.’ So only tentative structures of lignosulfonates were proposed (Glennie DW (1971) Reactions in sulfite pulping. In: Sarkanen KV, Ludwig CH (eds) Lignins; occurrence, formation, structure and reactions. Wiley-interscience, New York, Sydney, Toronto, pp. 607)

What is the additional modification to convert lignosulfate B to lignosulfate C

Please describe the additional purification mention in conversion or lignosulfate F to E.

(Response)

We thought you would like to point out the conversion of lignosulfonate D to E, not from lignosulfonate D to E.  We have changed the Materials and Methods as described below.

All lignosulfonates samples used in this study are provided by Nippon Paper Industries Co., Ltd. (Japan). Their preparative method, molecular weight (determined by gel permeation chromatography) and purity are shown in Figure 2.  Lignosulfonate A is derived from a sulfite spent liquor and contains sodium lignosulfonate as a main component. Lignosulfonate B and Lignosulfonate C are both purified and high molecular weight sodium lignosulfonates, and latter one applied some additional modifications including oxidation treatment for its preparation. Lignosulfonate D is a purified and partly desulfonated sodium lignosulfonate produced with alkaline condition at high temperature and pressure, and then purification. Lignosulfonate E is a similar type as Lignosulfonate D but having a higher purity of lignosulfonate produced with an enhanced purification including additional washing process to remove impure substances. Lignosulfonate F is also a similar type as Lignosulfonate D. Lignosulfonate F is a weakly acidic product although Lignosulfonate D is a weakly alkaline one, and the water-solubility of Lignosulfonate F is lower than that of Lignosulfonate D.

Describe QOL (in line 328)

(Response)

We spelled out “quality of life”.

Other major corrections

Page 13, line 362:  “two or three fold higher than that of povidone iodine” corrected to “two or three order higher than that of povidone iodine”

Figure 6: E. “polyssacharide” was corrected to “polysaccharide”  

Figure 7  “Lignosufonate F” was corrected to “Lignosulfonate F”

Fourteen new references and one supplementary Table S1 were added.
